# Dental Care for Older Adults

**DOI:** 10.3390/ijerph20010214

**Published:** 2022-12-23

**Authors:** Katherine Chiu-Man Leung, Chun-Hung Chu

**Affiliations:** Faculty of Dentistry, The University of Hong Kong, Hong Kong, China

**Keywords:** older adult, masticatory function, xerostomia, medical conditions, replacement of missing teeth, silver diamine fluoride

## Abstract

There is a global increase in the older population. Unfortunately, dental conditions in the older population can sometimes be poor as a result of worsened physical conditions and the cumulative damage caused by dental diseases in the past. Many suffer from oral diseases such as dental caries and periodontal disease but receive no regular dental care. Oral conditions and systemic problems are interrelated. Chronic medical problems and polypharmacy are common among them. These conditions may lead to xerostomia with or without a decrease in saliva output. Additionally, many older adults have deteriorated masticatory function associated with physical health issues such as frailty. Preventive measures are crucial to stop oral diseases from progressing and the replacement of missing teeth is needed when masticatory function is impaired. Older adults also suffer a higher risk of oral cancer because of their less resilient but more permeable oral mucosa. With the increasing need for elderly dental care, dentists should equip themselves with knowledge and skills in geriatric dentistry. They should help older adults to develop and maintain the functional ability that enables well-being in older age. This communication article aims to discuss the relevant medical conditions, common dental diseases, and dental care for older adults.

## 1. Introduction

People today tend to live longer and retain more teeth into later years of life as a result of improvements in living standards, advancements in medical technology, and improved accessibility to medical and dental care. The population of older people in the world is expected to increase from 0.7 billion in 2019 to 1.5 billion in 2050 [1]. Unfortunately, the dental condition of many older adults is poor due to worsened physical conditions and cumulative damage caused by dental diseases in the past. Furthermore, chronic non-communicable diseases are common among them. Degenerative changes, as well as chronic diseases and their treatments, can negatively affect oral health.

Medical and dental conditions are interrelated. Some chronic systemic diseases have a direct effect on oral tissues. The side effects brought about by the treatment of these diseases can adversely affect oral health. Likewise, some oral diseases may affect the medical condition of older adults. Dentists and other healthcare workers need to have a sound understanding of the age-related physiological changes in oral tissues, the common medical conditions and their dental effects, and the relationship between oral and systemic diseases, as well as the intricacies of change related to ageing, in order to provide safe and effective dental care for medically compromised older patients.

Considering these degenerative changes and the impacts of the underlying systemic diseases, dentists should collaborate with other health professionals and be prepared to modify their treatment approach according to the elders’ needs and their health conditions. An approach to dental care in older adults needs to take into consideration their medical conditions and self-care ability and must prepare for the anticipated further deterioration of general health and ability when they advance in age. 

Healthy ageing is defined as the process of developing and maintaining the functional ability that enables well-being in older age [2]. It is about creating the environments and opportunities that enable people to be and do what they value throughout their lives. The WHO emphasizes that being free of disease or infirmity is not a requirement for healthy ageing, as many older adults have one or more health conditions that have little influence on their well-being when they are well-controlled [3]. One important feature of healthy ageing is to maintain a functional ability that meets a person’s basic needs. Apart from aesthetics, the main function of dentition is masticatory. Hence, in dentistry, healthy ageing can be achieved by the control of dental diseases and the maintenance of masticatory function. This article aims to discuss medical conditions, common dental diseases, and dental care for older adults.

## 2. Medication Conditions and Medications

Many older adults suffer from diabetes mellitus (DM), a common endocrine disorder that has been listed as one of the top ten causes of death in 2019 [1]. A review elaborated on the association between DM and dental problems, as well as conditions such as periodontal disease, delayed wound healing, taste alteration, and dental infections [4]. Diabetic patients have a high risk of periodontal disease, and their periodontal condition can be worsened if their diabetes is poorly controlled. Another review suggested a bidirectional relationship between DM and periodontal disease [5]. The risk of periodontitis increases with uncontrolled DM while severe periodontitis adversely affects glycaemic control. Salivary gland dysfunction causes decreased salivary flow in diabetic patients, which is explained by the hyperglycemia, hyperinsulinemia, and dyslipidemia that result in increased oxidative stress, inflammation, increased sympathetic activity, and impaired insulin signalling in the salivary glands [6]. Furthermore, delayed wound healing is often observed in DM patients and is primarily related to suppressed osteoblastic activity and reduced bone formation potential. Diabetic patients are more prone to dental infections such as stomatitis which may or may not be related to Candida infection. Denture-wearers with DM are also more susceptible to traumatic ulcers of the oral mucosa of the denture-bearing area than non-DM denture-wearers, probably due to slower healing or delayed wound repair [7]. Older adults with DM should maintain good oral hygiene not only for preventing periodontal disease but also for better glycaemic control. Regular dental visits for denture maintenance to avoid denture trauma are highly recommended.

Stroke and dementia can lead to deterioration in self-care ability, resulting in the need for assistance in carrying out basic daily living activities. Studies have reported that people with dementia often present with poor oral hygiene, gingival bleeding, periodontal pockets, mucosal lesions, and reduced salivary flow [8]. Apart from increased dental plaque accumulation, poorer periodontal health, and infection of the oral mucosa, stroke survivors also show impairment in mastication and swallowing, which limits their food intake [9].

Saliva is important for maintaining oral health. It facilitates chewing, lubricating, tasting, cleansing, and speaking. Its mineralising, buffering, and antimicrobial properties are crucial for the prevention of dental caries and resistance to dental infections. Both major and minor salivary glands undergo degenerative changes with age. Diseases such as Sjögren’s syndrome [10], and an array of medications including antidepressants and some diuretics, are known to reduce saliva secretion. Their long-term use can increase one’s susceptibility to caries and periodontal diseases. One simple clinical assessment to detect oral dryness is to test if the oral mucosa sticks to the dental mirror. If a reduction in saliva flow is suspected, measurement of the salivary flow rate is warranted. Unstimulated whole salivary flow, the normal rate of which is about 0.3–0.4 mL/min, can be measured with simple equipment.

Xerostomia, the subjective feeling of dry mouth, is a common complaint among older adults. A systematic review concluded that up to one third of older adults have xerostomia [11]. Nonetheless, xerostomia is not often presented as the chief complaint. Sometimes complaints of xerostomia may be subtle and indirect. For instance, complaints of choking when dry food is eaten, dry cough, and the tongue sticking to removable dentures can be the first signs of xerostomia. Older adults taking medications such as sedatives and antihypertensive drugs are at high risk of salivary gland hypofunction [11]. Furthermore, polypharmacy is a common phenomenon in older adults [12]. The more drugs taken, the higher the prevalence of xerostomia. It is prudent to ask about the number of drugs an older adult is taking besides taking the disease count. People with xerostomia are at risk of dental caries and periodontal disease. They may also experience difficulties in speaking and swallowing, burning mouth syndrome, and taste alteration. Some patients may develop habits of consuming acidic food and drinks to stimulate salivary flow. Such habits can lead to severe tooth surface loss due to erosion, which jeopardizes aesthetics when the anterior teeth become shortened. This can pose difficulties for oral rehabilitation because of reduced crown height and lack of interocclusal space. Dentists, dental hygienists, and other oral healthcare workers should provide tailored oral hygiene instruction and dietary advice, as well as regular fluoride applications, for the prevention of dental caries. Saliva substitutes or hydrating mouthwash can be prescribed to relieve xerostomia symptoms. Saliva stimulants, such as cevimeline, which is a selective acetylcholine analogue with a high affinity for the M3 receptor, have been reported to improve the oral health–related quality of life of Sjögren’s syndrome patients [13].

## 3. Dental Diseases and Their Management

### 3.1. Dental Caries

Dental caries is a major non-communicable disease affecting the vast majority of older adults. The incidence of coronal and root caries increases with age [14]. The annual increments of coronal and root caries have been estimated to be 0.86 [15] and 0.5 [16] surfaces, respectively. A recent systematic review highlighted that the trend of caries had shifted from children to adults with the third peak of caries emerging at around the age of 70, due to the appearance of root caries [17]. Root caries is not only expensive to treat but also not easy to restore because the carious lesions are often located at sites with difficult access. Moisture control may be difficult and hence proper bonding of restorative material may be compromised. In fact, a high failure rate of restoring root caries with glass ionomer cements and composite resin has been reported [18]. Moreover, caries can damage the tooth and the restorative procedures can further weaken it, risking its fracture upon occlusal loading and making it unsuitable as an abutment to support a dental prosthesis.

It has been reported that people who are older, tobacco users, and those with more gingival recession and poorer oral hygiene have a higher risk of root caries [19]. Similarly, coronal caries experience, maxillary teeth, buccal root surfaces, gingival recession, and plaque on the root surface have been identified as risk indicators of root caries in users of long-term care facilities [20]. Clinicians should be well aware of these predisposing factors and identify people who are prone to root caries so that individualized programmes for effective prevention and management of root caries can be designed and carried out.

Fluoride is an effective anti-caries agent which halts demineralization and promotes remineralization of enamel and dentine [21]. To be effective, it has to be in contact with tooth surfaces. Daily toothbrushing using fluoridated toothpaste as a delivery mode of fluoride is the most commonly used method. Most fluoridated toothpaste for adult use contains 1000–1450 ppm fluoride. Since older people are considered a high-risk group for caries, a prescription of high-fluoride toothpaste containing 5000 ppm fluoride can be considered. High-fluoride toothpaste is more effective than toothpaste with a lower fluoride concentration in the improvement of surface hardness of untreated root caries lesions [22] and in preventing and arresting root carious lesions [23].

Silver diamine fluoride (SDF) has been extensively researched in the last two decades for its use in caries management in older adults. Silver exerts an anti-bacterial effect against cariogenic bacteria while fluoride promotes remineralization. It is effective in the prevention and arrest of root caries, the remineralization of deep occlusal lesions, and the treatment of hypersensitive dentine [24]. Among the professionally applied topical fluorides, an annual application of 38% SDF solution combined with oral health education has been shown as the most effective way of preventing dental root caries [22]. However, it should be noted that the major drawback of using SDF is the unsightly black stain left on the lesions due to the formation of silver products. In addition, the bond strength of restorative material such as glass ionomer cements may be negatively affected in cavities previously treated with SDF [25]. Alternatively, quarterly professionally applied 5% fluoride varnish and oral hygiene instructions have been shown to be equally effective and can be used in places where SDF is not available or in people who cannot accept the side effects of SDF [26].

### 3.2. Periodontal Disease

Periodontal disease is a chronic inflammatory disease induced by dental biofilm. It causes damage to the supporting structures which can result in gingival recession, alveolar bone resorption, tooth mobility, and eventual tooth loss. Periodontal disease affects over 60% of older adults, possibly due to decreased immunity, poor manual dexterity, and visual degeneration or impairment in carrying out proper oral hygiene practices [27]. Gingival recession due to periodontal disease exposes root surfaces that are previously unexposed. These exposed root surfaces are prone to caries attack if plaque accumulation persists.

Teeth may become mobile when the extent of attachment loss is such that the teeth are not well supported. This not only negatively affects mastication efficiency but also leads to discomfort during chewing. When tooth replacement is considered, mobile teeth are poor abutments to support dental prostheses. A more complex removable partial denture design may be necessary to prepare for the future loss of periodontally-involved teeth with a doubtful prognosis. Taking impressions is challenging. On the one hand, mobile teeth need to be recorded without any displacement, which can be difficult when they show lateral and vertical mobility; on the other hand, retrieving the impression may cause discomfort or worst still accidental extraction of severely mobile teeth. Blocking out the undercut around mobile teeth is necessary to avoid these problems but the contour of these teeth cannot be accurately captured.

A supportive periodontal therapy regimen that includes the improvement of oral hygiene and removal of barriers for easy access to oral hygiene tools is mandatory. Extraction of markedly mobile teeth and teeth with poor prognosis ought to be considered so that oral hygiene efforts can be focused on the relatively healthy remaining dentition. Regarding mobile mandibular anterior teeth in patients with severe periodontitis, the use of splints, using composite resin with or without stainless steel wires, during or after supportive periodontal therapy has been advocated. The survival rate of splints until fracture or debonding was reported as 74.4% after 3 years, and the reduction in periodontal probing depth and clinical attachment loss remained stable [28]. Nonetheless, it is not uncommon to see splints fractures after only several months of placement when teeth with great differential mobility are splinted together. Careful selection for splinting is warranted.

Having good oral hygiene practices can improve oral health and prevent periodontal disease. However, older people often present with inadequate oral hygiene. Low awareness, low priority, and physical constraints are commonly claimed to prevent them from carrying out proper oral hygiene practices. For instance, sarcopenia makes the physical act of toothbrushing more challenging [29]. In people with deformed hand joints due to rheumatoid arthritis, proper gripping of a toothbrush can be difficult. Modifying the toothbrush with a foam handle may help patients to obtain a better-controlled grip so that they can carry out proper toothbrushing. Alternatively, electric toothbrushes can be recommended. Similarly, interdental cleaning with dental floss can be replaced by a water flosser and modified interdental brushes [30].

Caregivers who assist frail elders to carry out basic daily activities should be educated and trained in the provision of oral hygiene practices, especially those who provide care for institutionalized elders. A 5S strategy originally designed for the organization of space to enhance work efficiency has been advocated in dentistry to improve the oral hygiene practice of patients with Alzheimer’s disease, with an aim to avoid waste of time and risk of injury [31]. 5S comes from Japanese words which mean sorting, setting in order, shining, standardising, and sustaining. Following the 5S checklist of sustained oral hygiene practices, together with the empowerment of caregivers to guide and monitor the process, patients can be expected to develop a regular and effective oral hygiene practice. Additionally, a cohort study showed that outreach dental programmes can be helpful in solving some of the dental problems of older adults using long-term care services and protecting most of them from deterioration in oral health–related quality of life, especially those suffering from toothache [32].

## 4. Masticatory Function and Xerostomia

Masticatory function is an important attribute of overall health. The ability to chew properly has been associated with physical health factors including frailty and sarcopenia [33], cognition status [34], high-level functional ability, such as intellectual activities and social roles [35], and life quality [36] in older adults. People with deteriorated masticatory function tend to consume a softer diet and alter their food with less fibrous content and an increased intake of carbohydrates and fat [37]. Such a diet negatively affects the nutritional status and general health of individuals and has been identified as a high risk factor for many chronic diseases including atherosclerosis and cancer [38]. Apart from low fibrous content, such a diet does not require extensive chewing, which in turn negatively affects the strength of the masticatory muscles. The frequency of meals may also be increased with in-between meal snacking habits. As dietary fibre helps to cleanse the teeth during mastication, teeth are less well-cleansed if the diet is low in fibre. Moreover, the action of chewing can stimulate salivary flow where saliva exerts an important protective effect on the oral cavity through its flow and composition such as immunoglobulins. A higher proportion of carbohydrates in these diets and increased meal frequency suggest a prolonged substrate supply for cariogenic bacteria, increasing the risk of caries.

Training of the oral muscles has been recommended to improve oral function in older adults. These exercises include (1) chewing exercises where chewing gum or other chewing materials are used for the practice of chewing twice a day for 5–10 min each time, (2) clenching exercises where individuals repeatedly clench and release, and bite and hold, using a chewing aid apparatus, and (3) simple oral exercises where stretching of masticatory muscle and the tongue is performed. A recent systematic review concluded that oral exercises can improve the maximum bite force but the effectiveness on masticatory performance was inconclusive [39]. Among the various types of oral exercises, chewing exercise is the most effective, followed by clenching exercise, but simple oral exercise may not be effective. Other studies also found increased unstimulated salvia secretion and swallowing function after oral exercises [40].

## 5. Edentulism and Replacement of Missing Teeth

Tooth loss is the endpoint of severe dental caries and periodontal disease. After tooth extraction, teeth adjacent to the extraction site may drift towards each other and the opposing tooth may over-erupt. Loss of teeth can adversely affect aesthetics, speech, and chewing function. Tooth loss resulting in edentulism also affects nutritional intake, which in the long term can cause malnutrition. Moreover, tooth loss has a negative impact on social status, self-esteem, and oral health–related quality of life [41]. Older adults with multiple missing teeth have a higher risk of dementia than those with more teeth [42].

Many older adults have lost all their molar teeth due to dental diseases and are left with premolars and the anterior teeth only. This compromised dentition is often referred to as shortened dental arch (SDA). The SDA concept can be a pragmatic and useful treatment approach in older adults. It emphasizes a functional dentition and indicates that dental arches comprising the anterior and premolar regions meet the requirements of functional dentition [43]. SDA can last for 27 years or more and the treatment needed is found to be comparable to a complete dental arch [44]. In other words, the provision of the posterior occlusion in SDA situations may not be warranted. Reports have revealed that approximately a quarter of removable partial denture-wearers chose not to wear their dentures and preferred continued function with a severely shortened lower arch despite apparent limitations [45]. Furthermore, no improvement in the oral health–related quality of life has been shown in people with SDA who had their missing posterior teeth replaced, thus additional prosthodontic treatment for people with SDA may not be justified [46]. However, it must be noted that inadequate periodontal support of the remaining teeth is a contraindication of the SDA approach. There is some evidence to show that traumatic occlusal force, i.e., any occlusal force resulting in the injury of teeth and/or the periodontal attachment apparatus, may be associated with the severity of periodontitis [47]. Therefore, a complete periodontal assessment of the remaining teeth in partially edentulous patients is important before decisions regarding the replacement of missing teeth are made.

In situations where the replacement of missing teeth in older adults is necessary, complicated designs that have a high demand for manual dexterity for care should be avoided. In the past few decades, dental implants have been increasingly popular for tooth replacement as an expanded treatment option, and promising outcomes have been achieved. In short-bounded saddle situations, the implant-supported prosthesis has become the treatment of choice, replacing the conventional fixed partial dentures that used to be the first line of treatment. In edentulous people, an implant-supported mandibular complete overdenture based on two implants has been shown to be significantly more stable and comfortable to wear and achieves better chewing efficiency and speaking than a conventional complete denture. Hence, it has been recommended as the standard of care [48]. Nonetheless, not everyone can receive implants. A systematic review revealed that diabetic patients with dental implants exhibited more marginal bone loss than non-diabetic patients, albeit with no significant difference in the rate of implant failure [49]. Furthermore, dentists should be cautioned that if a person becomes cognitively impaired and care-resistant, implants may become a liability as they become a source of infection for aspiration pneumonia [50].

Retention and stability of removable dentures may deteriorate after years of wearing due to continuous resorption of the alveolar bone. The rate of bone resorption is most rapid in the first six months after tooth extraction, and the mandible can resorb up to five times as much as the maxilla [50]. Although the rate of bone resorption varies among individuals and is multifactorial, it is stated that anatomic, biologic, and mechanical factors are the major factors leading to the reduction of residual ridges [51]. Removable dentures with poor retention and stability can induce pain during mastication due to soft tissue trauma and lead to further bone resorption. Masticatory performance is also negatively affected. In general, it is advisable to replace the removable dentures after approximately five years of use [52].

Nonetheless, at times when a remake of the prosthesis cannot be arranged, or in xerostomic denture-wearers whose dentures are unretentive even when they are well-constructed, denture adhesive can be useful. Denture adhesives act by swelling and becoming viscous in the presence of water or saliva thus filling up the space between the denture base and the oral mucosa. It has been shown that denture adhesives can improve denture retention, bite force, and masticatory performance of complete denture–wearers [53].

## 6. Oral Cancers and Precancerous Lesions

Cancer of the oral cavity, tongue, lips, and oropharynx is one of the top ten common cancers in the world. The annual global incidence of oral cancer has been reported as approximately 500,000 [54]. The prevalence of oral cancers is higher among older adults, in particular those in underdeveloped countries. Older adults may have thin and non-resilient oral mucosa, which becomes permeable to toxic substances. The less disease-resistant mucosa may predispose older adults to cancerous and precancerous lesions, which can be serious and life-threatening. A systematic review reported a positive association between periodontal diseases and oral cancer [54]. The authors explained the association between similar inflammatory cells and mediators produced in response to periodontal diseases and the chronic inflammation induced by periodontal pathogens that promote the already initiated cells, leading to the breakdown of normal cell growth control and prospective carcinogenesis. A meta-analysis also reported that ill-fitting dentures substantially increased the risk of cancer development with an odds ratio of 3.9 [55]. Ill-fitting dentures induce chronic mucosal irritation that leads to trauma and precancerous lesions, which can develop into cancer. Surgical excision, radiotherapy, and chemotherapy are the usual treatments for oral cancer. Excision of the lesion and the associated structures often results in an extensive loss of hard and soft tissues that requires complex rehabilitative treatment. Trismus may result after radiotherapy because of muscle fibrosis. Reduction in salivary flow is common when the salivary glands are damaged. Strict oral care protocol, including rigorous oral hygiene practice and topical fluoride therapy, has to be followed to prevent dental caries, periodontal disease, and other oral infections.

## 7. Conclusions

In conclusion, dental care for older adults requires special attention to their physical and medical conditions. Although older adults are at risk of oral diseases, most of these diseases are preventable and hence prevention and early treatment are important. To achieve healthy ageing, dental treatment approaches should be simple and prevention-focused with an aim to stop oral diseases from progressing and restore masticatory function with teeth replacement.

## Data Availability

Not applicable.

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
