# Peer review of "Dental Care for Older Adults"

_ijerph, 2022, doi:10.3390/ijerph20010214_

Round 1

Reviewer 1 Report

This is a very well presented paper with a clear structure and style. 

A few minor points of note include: 

line 102 - use of the word alarming is unnecessary and emotive

line 113 - broaden the comments about OHE instruction to include the dental team, not just the dentist

line 134 - the comments about SES is interesting but needs further explaining. It is presented here in the same way as clinical risk factors. Given the journal focuses on public health details to elaborate this point would increase the relevance of the paper to the journal and to the readers. 

line  148 - use of the word 'regular' is incorrect. Regularity is a measure of frequency, and not appropriate here. Perhaps replace with '...lower fluoride content toothpastes...', or similar.

line 207 and 275 - incorrect use of the word 'shall'. This changes the tense of the writing to future tense. Perhaps replace with 'should'.

As mentioned these are minor points. However, a larger item to consider is not about the quality of this work, which I feel is good, but the appropriateness of the content to the audience. This is very much a clinical article written I would suggest with dental professionals, particularly dentists in mind. It would be useful to consider some of the points you raise about the oral health of older people and prevention of disease from a public health perspective. This would broaden the appeal of the paper and make it more relevant to the audience of this journal. 

Lastly, I want to commend you on the presentation of this work. It is by far the best written paper I have reviewed for a long time. Well done. 

Author Response

Point 1: line 102 - use of the word alarming is unnecessary and emotive

Response 1: line 102 - “are alarming” is changed to “can be the first”

Point 2: line 113 - broaden the comments about OHE instruction to include the dental team, not just the dentist

Response 2: Thank you for the advice. “Dental hygienists and other oral healthcare workers” is added.

Point 3: line 134 - the comments about SES is interesting but needs further explaining. It is presented here in the same way as clinical risk factors. Given the journal focuses on public health details to elaborate this point would increase the relevance of the paper to the journal and to the readers.

Response 3: Elaboration on SES is added.  People of lower socioeconomic status cannot afford routine dental care or dental treatment if they are not subsidized. Dental health may not be a priority for people who attain a low education level.

Point 4: line  148 - use of the word 'regular' is incorrect. Regularity is a measure of frequency, and not appropriate here. Perhaps replace with '...lower fluoride content toothpastes...', or similar.

Response 4: “regular fluoride toothpaste” is changed to “toothpaste with lower fluoride concentration”.

Point 5: line 207 and 275 - incorrect use of the word 'shall'. This changes the tense of the writing to future tense. Perhaps replace with 'should'.

Response 5: The sentence is changed as advised.

Point 6: As mentioned these are minor points. However, a larger item to consider is not about the quality of this work, which I feel is good, but the appropriateness of the content to the audience. This is very much a clinical article written I would suggest with dental professionals, particularly dentists in mind. It would be useful to consider some of the points you raise about the oral health of older people and prevention of disease from a public health perspective. This would broaden the appeal of the paper and make it more relevant to the audience of this journal. 

Response 6: Thank you for your comments. This paper is written mainly for dentally trained readers, such as dentists and other dental professionals. The reviewer rightly pointed out that it would be useful to consider some of the points from a public health perspective. However, the scope could be too wide and out of the expertise of the authors.

Point 7: Lastly, I want to commend you on the presentation of this work. It is by far the best written paper I have reviewed for a long time. Well done. 

Response 7: Thank you for your commendation. It is very encouraging.

Reviewer 2 Report

Dear authors,

The article entitled Dental care for older adults brings together useful and important information upon the most commonly met oral problems of the elderly and associated factors.

However, some important elements are missing from the paper.

First, there is no section of Materials and methods. A description of the searched databases, of the keywords used, of the time period considered, and of the inclusion and exclusion criteria are very important.

Another important section that is missing is the Discussion chapter, where comments upon the most important results of the study, the limitations of the study and some directions for future research can be made.

Finally, many other aspects concerning oral care of the elderly can be considered, such as oral health-related quality of life, socioeconomic aspects, preventive strategies targeting this age group, and community programs addressed to the elderly.

Author Response

Point 1: The article entitled Dental care for older adults brings together useful and important information upon the most commonly met oral problems of the elderly and associated factors

Response 1: Thank you for your comments.

Point 2: However, some important elements are missing from the paper. First, there is no section of Materials and methods. A description of the searched databases, of the keywords used, of the time period considered, and of the inclusion and exclusion criteria are very important.

Response 2: This is a communication article, rather than a systematic review or a cohort study. Therefore, there is no materials and methods section.

Point 3: Another important section that is missing is the Discussion chapter, where comments upon the most important results of the study, the limitations of the study and some directions for future research can be made.

Response 3: By the same token, there is no “discussion” of the results, limitations of the study, or future research directions presented.

Point 4: Finally, many other aspects concerning oral care of the elderly can be considered, such as oral health-related quality of life, socioeconomic aspects, preventive strategies targeting this age group, and community programs addressed to the elderly.

Response 4: This discussion article is of adequate length. We will discuss your suggested other aspects in our future papers. Thank you for your suggestions.

Round 2

Reviewer 1 Report

Thank you for amending this paper. 

Point 3, line 134 still remains somewhat simplistic and in its current format it comes across judgemental, even if this is not the intention. Given the readership of this journal more care should be taken on this point to elaborate the public health elements related to oral health. 

Overall a well presented paper. 

Author Response

Thank you for your comments. The phrase “lower socio-economic status” is deleted.

Reviewer 2 Report

Dear Authors,

According to the data available on mdpi site, Communication type articles "present groundbreaking preliminary results or significant findings that are part of a larger study over multiple years" or "include cutting-edge methods or experiments, and the development of new technology or materials", which is not the case of the present study. Also, according to the site, "the structure is similar to an article"(MDPI | Article Types)

For all these reasons, I consider that all the observations I made in the first evaluation report remain valid.

Given the fact that the article contains useful information and is written in a manner that is easy for readers to follow, please allow me to suggest placing the article in the "narrative review" category and completing it with the necessary data for publication.

Author Response

Thank you for your suggestion. “Narrative review” does not seem to be an option in the list of article types. As this article is prepared as a communication rather than a systematic review or a cohort study, we are afraid there is no materials and methods section, results, discussion of the results etc.